# Antioxidant and Anti-Diabetic Activities of Polysaccharides from Guava Leaves

**DOI:** 10.3390/molecules24071343

**Published:** 2019-04-05

**Authors:** You Luo, Bin Peng, Weiqian Wei, Xiaofei Tian, Zhenqiang Wu

**Affiliations:** 1School of Biology and Biological Engineering, South China University of Technology, Guangzhou 510006, China; 1842you@sina.cn (Y.L.); pengbin163163@163.com (B.P.); 201720143798@mail.scut.edu.cn (W.W.); xtien@scut.edu.cn (X.T.); 2Pan Asia (Jiangmen) Institute of Biological Engineering and Health, Jiangmen 529080, China

**Keywords:** polysaccharides, guava leaves, antioxidant, anti-diabetic

## Abstract

Guava (*Psidium guajava* L., Myrtaceae) leaves have been used as a folk herbal tea to treat diabetes for a long time in Asia and North America. In this study, we isolated polysaccharides from guava leaves (GLP), and evaluated its antioxidant activity in vitro and anti-diabetic effects on diabetic mice induced by streptozotocin combined with high-fat diet. The results indicated that GLP exhibited good DPPH, OH, and ABTS free-radical scavenging abilities, and significantly lowered fasting blood sugar, total cholesterol, total triglycerides, glycated serum protein, creatinine, and malonaldehyde. Meanwhile, it significantly increased the total antioxidant activity and superoxide dismutase (SOD) enzyme activity in diabetic mice, as well as ameliorated the damage of liver, kidney, and pancreas. Thus, polysaccharides from guava leaves could be explored as a potential antioxidant or anti-diabetic agents for functional foods or complementary medicine.

## 1. Introduction

Diabetes mellitus (DM) is the most common progressive disease which is characterized as continuous hyperglycemia due to the impairment of insulin production by pancreatic β-cells and/or caused by peripheral insulin resistance [1]. Long-term hyperglycemia is associated to increasing dyslipidemia, reactive oxygen species production, and declining antioxidant status [2]. Oxidative stress is one of the main mechanisms of progression of diabetes and actively leads to cellular damage that precedes the onset of many diabetic complications [3].

Oral hypoglycemic drugs sold on the market have side effects such as gastrointestinal discomfort, weight gain, and hepatic dysfunction [4]. Therefore, it is an urgent need to find new potential agents for prevent and treat DM. Plants are known to possess a wide variety of pharmacological effects and extraordinary therapeutic possibilities. *Psidium guajava* Linn. is a common fruit plant available in many countries having tropical and subtropical climates. Guava leaves have been used as a folk medicine or herbal tea to treat diarrhea [5] and diabetes [6,7] in India, China, Pakistan, Bangladesh, and Mexico for a long time due to lower toxicity and good therapeutic function [8,9]. Díaz-De-Cerio et al. verified that the hypoglycemic effects of guava-leaf ethanolic extract were associated with improving endothelial dysfunction in obesity mice [10]. Shen et al. verified that aqueous soluble extract from guava leaves has antihyperglycemic function against type 2 diabetes [11]. Previous studies were focused on flavonoids and phenolic compounds extracted from guava leaves. However, no study has been conducted to prove the anti-hyperglycemic effect of polysaccharides from guava leaves. Polysaccharides from natural products have become a research hotspot because of their multiple biological activities such as antioxidant [12,13], anti-inflammatory [14], anti-diabetic [15,16,17], immunomodulatory [18,19], and anti-tumor [20,21] effects.

This study thus aims to demonstrate the anti-diabetic activity of polysaccharides from guava leaves using the mice model induced by streptozotocin combined with a high-fat diet.

## 2. Results and Discussion

### 2.1. The Composition and Characteristics of Guava Leaves (GLP)

The content of total sugar in GLP was 62.58% without reducing sugar, and the content of uronic acid was 7.59%. It meant that neutral polysaccharides were dominant. GLP showed negative response to coomassie brilliant blue reaction. It indicated that there was no protein in GLP. Four fractions could be detected in GLP using HPLC-RI system and their retention time were 8.587 min, 10.57 min, 11.92 min, and 17.94 min, respectively (Figure 1). The proportion of four fractions was about 23.28%, 7.57%, 3.04%, and 66.10%, respectively. Their molecular masses were 957.09 kDa, 288.40 kDa, 127.40 kDa, and 3.34 kDa, respectively. It illustrated that the majority of GLPs were low molecular weight polysaccharides, followed by high molecular weight ones. Yuan et al. reported that polysaccharides extracted from mulberry leaves (MLP) had several fractions composed of high molecular weight polysaccharides (≥80.99 kDa) and low molecular weight polysaccharides (3.64 kDa) [13].

### 2.2. Antioxidant Activity of GLP

Free radicals in the body regulate cell growth as well as inhibit viruses and bacteria [22]. However, excessive free radicals will cause several chronic human diseases such as aging, cancer, and arteriosclerosis [23]. The antioxidant capacity of GLP was determined on the basis of their scavenging ability of DPPH, OH, and ABTS free radicals as shown in Table 1. The results indicated that GLP had strong abilities of scavenging DPPH, OH, and ABTS free radical with IC_50_ of 46.49 μg/mL, 175.52 μg/mL, and 102.82 μg/mL, respectively, which were all higher than that of the positive control, such as ascorbic acid or Trolox.

Many researchers also reported the antioxidant activity of polysaccharides from other natural products. Polysaccharides from maca leaves had strong effects on scavenging DPPH radical with an IC_50_ of 0.82 mg/mL [24]. Polysaccharides from olive leaves also displayed good ability of scavenging DPPH radical (IC_50_ = 34.80 μg/mL) [25]. Algal polysaccharides had been verified to play a crucial role as free radicals scavengers in vitro [26]. It suggested that the potential antioxidant activity of GLP might be better than that of some other similar products.

### 2.3. Hypoglycemic Activity of GLP

#### 2.3.1. Ameliorating Body Weight Loss

STZ could cause a severe loss in body weight since muscle destruction or degradation of structural proteins [27]. The body weight of model group (MG) mice declined obviously after STZ injection when compared with that of normal group (NG) mice (Figure 2). The body weight of MG mice continuously decreased from 31.07 g to 28.36 g. When treated with low-dose polysaccharides or high-dose polysaccharides for 2 weeks, the body weight of those diabetic mice was partly recovered. The results indicated that GLP could significantly ameliorate body weight loss, which was better than that of positive group (PG) (*p* < 0.05).

#### 2.3.2. Regulating Fasting Blood Glucose

After STZ injection, fasting blood glucose (FBG) levels in MG mice (14.37 mmol/L) were significantly higher than those of NG mice (3.44 mmol/L) (*p* < 0.01). In contrast, FBG levels of the mice treated with polysaccharides and acarbose significantly decreased (Figure 3). There was no significant difference of FBG levels between high-dose polysaccharides (HP) group (11.69 mmol/L) and PG group (10.40 mmol/L) (*p* > 0.05). The results indicated that GLP could effectively decrease FBG of diabetic model animals. The similar conclusion could be found in other plant polysaccharide research. Kiho et al. reported that acidic polysaccharide from *tremella aurantia* depressed the increase of plasma glucose in diabetes using genetically non-insulin-dependent diabetic model mice [28]. Jiao et al. demonstrated that polysaccharides from *Morus alba* fruit significantly reduced the FBG of type 2 diabetic rats induced by high-fat diet combined with streptozotocin injection [29].

#### 2.3.3. Regulating Biochemical Indictors

Total cholesterol (TC) is a key index of blood lipid in clinical practice which is defined as the sum of all lipoprotein cholesterol in the blood. High level of TC indicates high risks of atherosclerosis, coronary heart disease, and diabetes [30]. Triglycerides (TG) are the main constituents of body fat in humans, and excessive TG can cause fatty liver, obesity, and pancreatitis [31]. The results showed that MG mice had significant high levels of TC and TG compared with NG mice (*p* < 0.01). The acarbose and polysaccharides noticeably reduced the high levels of TC and TG of MG mice (*p* < 0.01) (Table 2).

The excretion of creatinine is an indicator for monitoring the kidney metabolism. The creatinine in MG mice increased about 2.4 times (*p* < 0.01) to that in NG mice. The content of creatinine in low-dose polysaccharides (LP) and HP mice decreased by 36.99% and 41.03%, respectively. It illustrated that polysaccharides effectively decreased creatinine content in MG mice. Accumulation of the blood creatinine occurs with the impaired renal function [32]. Therefore, it could be inferred that GLP has the activity of protecting kidney.

Glycated serum protein (GSP) level is a good indicator to reflect the average level of blood glucose in the past 1–2 weeks [33]. GSP level of MG mice (3.07 ± 0.15 mM) was much higher than that of NG mice (1.99 ± 0.11 mM) (*p* < 0.01). Acarbose and polysaccharides significantly reduced GSP level of MG mice in a dose-dependent way (*p* < 0.01). Compared with the MG mice, the GSP level in PG, LP, and HP mice decreased by 19.22%, 7.17%, and 20.85%, respectively. It was consistent with the FBG test result.

The antioxidant capacity was closely related to body health. A reduced antioxidant capacity could easily cause inflammation, cancer, diabetes, and other disease [34]. The total antioxidant capacity of MG mice was much lower than that of NG mice (*p* < 0.01). However, polysaccharides significantly enhanced the total antioxidant capacity of MG mice in a dose-dependent way (*p* < 0.01). The total antioxidant capacity of LP and HP was 0.70 mM and 0.75 mM, respectively.

SOD enzyme plays a significant role in enzymatic defense system. SOD activity in diabetes model mice significantly decreased when compared with that in normal mice (*p* < 0.01). When diabetic mice were treated with acarbose and polysaccharides, their SOD activity significantly improved (*p* < 0.01). Total superoxide dismutase (T-SOD) activity of PG, LP, and HP mice was 721.83 U/mgprot, 702.13 U/mgprot, and 730.80 U/mgprot, respectively.

Lipid peroxidation is cytotoxicity by forming malondialdehyde, which can cause cross-linking polymerization of proteins, nucleic acids, and other macromolecules. Nutrient overload such as hyperglycemia and hyperlipidemia could stimulate the lipid peroxidation and generate α, β-unsaturated 4-hydroxyalkenals [35]. Malonaldehyde (MDA) content in MG mice was much higher than that in NG mice (*p* < 0.01). When compared to MG mice, acarbose and polysaccharides effectively reduced MDA content, and MDA content in PG, LP, and HP decreased by 27.12%, 22%, and 28.08%, respectively.

GLP had strong free radical scavenging capacities in vitro and also enhanced the antioxidant status in diabetic mice. The results are similar to the previous studies. Polysaccharides from Fuzhuan brick teas displayed good free radical scavenging activity in vitro and also had protective effects on high-fat diet-induced oxidative injury in vivo [36]. Polysaccharides from the algae *Gracilaria caudata* exhibited significant antioxidant activity in vitro and also greatly improved the antioxidant system in rats [37]. Oxidative stress is produced under diabetic conditions in various tissues and damages cellular organelles, which increases lipid peroxidation and causes insulin resistance [38]. Therefore, prevention of oxidative stress may be a potential method to avoid type 2 diabetes. Many studies verified that the anti-diabetic activity of polysaccharides in part resulted from their antioxidant effects [39,40].

#### 2.3.4. Protective Effect on Liver, Kidney, and Pancreas

Significant differences in the pattern and number of mice hepatocytes were observed among different experimental groups, shown in Figure 4. Normal hepatic cells had clear boundaries and round nucleus, which was surrounded by rich cytoplasm (Figure 4a). The inflammatory cells infiltration, cell swelling, focal necrosis and plasma osteoporosis, translucent, but still in the shape of polygons appeared in MG mice (Figure 4b). In the acarbose group (PG), changes in size and shape of hepatic cells were not evident, but inflammatory cells infiltration and some cytoplasm transparent occurred (Figure 4c). HP apparently effectively alleviated the symptoms of focal necrosis and infiltration of lymphocytes in the MG mice (Figure 4e).

The H&E stained sections of the renal tissue samples are presented in Figure 5. The regular shape of glomerulus, renal tubule and collecting duct was clear and distinguishable in the NG mice. Moreover, plump cells were neatly arranged, with little intercellular space. No abnormal symptoms were observed in kidney tissue (Figure 5a). In contrast, the renal cortex and medulla of the STZ-induced diabetic MG mice showed varying degrees of atrophy. Inflammatory cell infiltration and congestion in central vein indicated a poor condition. Glomerulosclerosis and irregular distribution of renal cells were observed (Figure 5b). Acarbose and polysaccharides relieved the kidney injury in diabetic mice to a certain extent. The shape and distribution of cells looked normal, but glomerular tissue was still infiltrated by the inflammatory cells (Figure 5c–e). HP showed better effects in terms of the protection of renal impairment (Figure 5e).

The protective effect on pancreatic islets is shown in Figure 6. The normal islet cells were closely arranged with clear boundaries (Figure 6a). However, serious pathological damages such as focal necrosis, inflammatory cell infiltration, and congestion in central vein were clearly observed in the MG mice (Figure 6b). Acarbose could repair the damage incurred by pancreas exposure to STZ. No lager area of cell necrosis occurred in the pancreatic tissue of PG mice, but vacuous areas and inflammatory cells can be observed (Figure 6c). The groups fed with LP (Figure 6d) and HP (Figure 6e) showed good effects on the recovery of impairment of pancreatic islets induced by STZ. The pancreatic cells were relatively homogeneous and arranged in good shape. Moreover, the focal necrosis and infiltration of inflammatory cell were improved evidently. The results were in consistent with protective effects of herbal polysaccharides on the pancreatic tissue of T2DM rats [29].

Histological analysis suggested that GLP effectively alleviated inflammation and protected the tissue structure of the liver, kidney, and pancreas in diabetic mice. This would help to control the deterioration of diabetes and the occurrence of related complications.

## 3. Materials and Methods

### 3.1. Materials and Chemicals

Guava leaves were obtained from Jiangmen Nanyue Guava farmer cooperatives (Guangdong, China). Leaves were dried at 60 °C, then pulverized and sieved (40 mesh) for the experiments.

1,1-Diphenyl-2-picrylhydrazyl (DPPH) was purchased from Shanghai Macklin Biochemical Co., Ltd. ABTS^+^ and Trolox were obtained from Aladdin Industrial Corporation (Shanghai, China). Ascorbic acid was purchased from Sinopharm chemical reagent Co., Ltd. (Shanghai, China). Dextran T-2000, T-500, T-70, T-40, and T-10 were purchased from Solarbio (Beijing, China). Acorbose was obtained from Sigma-Aldrich Co., Ltd. (St. Louis, MO, USA). Streptozocin was obtained from MP Biomedicals (Santa Ana, CA, USA). Assay kits for total cholesterol (TC), total triglycerides (TG), glycated serum protein (GSP), creatinine (CRE), total antioxidant capacity (T-AOC), total superoxide dismutase (T-SOD), and malonaldehyde (MDA) were purchased from Nanjing jiancheng Bioengineering Institute (Nanjing, China). The high fat diet consisted of 10% lard, 20% sucrose, 10% yolk powder, 0.5% sodium cholate, and 59.5% conventional feed, which was purchased from Jiangsu synergetic pharmaceutical bioengineering Co., Ltd. (Nanjing, China).

### 3.2. Extraction of Polysaccharides from Guava Leaves (GLP)

Each 10 g of dried guava leaves sample was pre-treated with 40 mL 95% ethanol to remove most of the polyphenols, pigments, monosaccharides, and fats. After filtration, the hot air-dried residues were extracted with 100 mL distilled water in an AS20500ATH ultrasonic bath (400 W, 40 KHz, Tianjin Automatic Science Instrument Co., Ltd., Tianjin, China) at 60 °C for 20 min (2 times). The aqueous extracts were concentrated using a rotary evaporator and deproteinized following the Sevag method [41]. After the removal of the Sevag reagent, the extracts were precipitated with ethanol to a final concentration of 70% at 4 °C overnight. The precipitation was dissolved in purified water and dialyzed against distilled water for 48 h with dialysis bag (molecular weight cut-off, 3000 Da), and then concentrated and lyophilized.

### 3.3. Analysis of the Composition of Polysaccharides

The total sugar content was determined by the phenol–sulfuric acid method using glucose as a standard [42]. The content of uronic acid was measured by the *m*-hydroxydiphenyl method using galacturonic acid as a standard [43]. Reducing sugar content was determined by dinitrosalicylic acid method [44]. Protein content was measured with the coomassie brilliant blue reaction [45].

The average molecular weight distribution of GLP was determined using a high-performance liquid chromatography (HPLC, Waters e2695 Separations Module, USA) equipped with a TSK-gel column (G4000PWXL, TOSOH, Japan) and a RI detector. The HPLC analysis was performed at 35 °C with the flow rate of 1 mg/mL. The calibration of standard curve was regressed with the retention time against the logarithm of the average molecular weight of the dextran standards (T-2000, T-500, T-70, T-40, and T-10). The average molecular weight of GLP was calculated with the calibrated curve.

### 3.4. Antioxidant Activities

#### 3.4.1. DPPH Radical Scavenging Activity

DPPH radical scavenging assay was carried out by following a method reported by Shimada et al. [46]. DPPH methanol solution of 0.6 mM concentration was prepared. A volume of 200 μL of DPPH solution was added to 100 μL GLP solution. The solution was evenly mixed and reacted at room temperature in dark. After 30 min, the absorbance of the solution was measured by a microplate reader (PerkinElmer, Singapore) at 517 nm. The analysis was in triplicate. Ascorbic acid was used as a positive control. DPPH radical scavenging activity was calculated by Equation (1).
(1)$$\mathrm{DPPH}\;\mathrm{radical}\;\mathrm{scavenging}\;\mathrm{rate}\;\left(\%\right)=\frac{1-\left(\mathrm{As}-\mathrm{Ao}\right)}{\mathrm{Ab}}\times 100$$
where As and Ao are the absorbance of the sample reaction solution and the sample solution without DPPH, and Ab is the absorbance of DPPH solution.

#### 3.4.2. OH Radical Scavenging Activity

OH radical scavenging activity was determined by salicylic acid method [47]. A volume of 500 μL sample solution was taken into a test tube. In total, 500 μL of 6 mM FeSO_4_ solution, 500 μL of 6 mM salicylic acid ethanol solution, 2 mL distilled water, and 500 μL of 2.4 mM H_2_O_2_ were successively added into the test tube. After incubation for 15 min at 37 °C in the water bath, the absorbance of the mixture solution was measured at 510 nm. Ascorbic acid was used as the positive control. Each sample was done for three times. Hydroxyl radical clearance rate was calculated by Equation (2).
(2)$$\mathrm{OH}\;\mathrm{radical}\;\mathrm{clearance}\;\mathrm{rate}\;(\%)\;=\frac{\mathrm{Ao}-\left(\mathrm{As}-A\right)}{\mathrm{Ao}}\times 100$$
where Ao represents the absorbance of distilled water (blank control), As is the absorbance value of sample solution, and A is the absorbance of sample solution without H_2_O_2_.

#### 3.4.3. ABTS Radical Scavenging Activity

The ABTS radical scavenging activity was evaluated by ABTS radical cation decolorization assay [48] with modifications. ABTS^+^ was prepared by mixing 7 mM ABTS^+^ solution with 2.45 mM K_2_S_2_O_8_ solution (1:1, *v/v*) at room temperature in dark for 16 h. The ABTS^+^ solution was diluted with purified water to an absorbance of 0.070 ± 0.02 at 734 nm. In total, 50 μL sample solution was reacted with 200 μL of ABTS^+^ at room temperature for 10 min in dark. The absorbance of the mixture was measured at 734 nm. Trolox was used as a positive control. The scavenging rate was calculated by Equation (3).
(3)$$\mathrm{ABTS}\;\mathrm{radical}\;\mathrm{scavenging}\;\mathrm{rate}\;\left(\%\right)=(1-\frac{\mathrm{As}}{\mathrm{Ab}})\times 100$$
where As is the absorbance of the sample and Ab is the absorbance value of the blank.

### 3.5. Animal Experiments

#### 3.5.1. Modeling and Drug Administration

Male ICR mice weighing 18–22 g were purchased from Hunan SJA Laboratory Animal Co., Ltd. (Changsha, China). Mice were raised in SPF-level lab (temperature of 23 ± 2 °C, a 12 h-light/12 h-dark cycle) and acclimatized in cages for 7 days.

After a week of adaptation, 8 mice were randomly selected as the normal group (NG). The NG mice were provided with conventional feed, while other mice were fed with high-fat diets for 3 weeks. After fasting for 12 h, the NG mice were intraperitoneally injected with 0.2 mL saline, whereas the other mice were injected with 0.2 mL of STZ solution at a dose of 40 mg/kg body weight. This treatment was repeated after two days. The fasting blood glucose (FBG) of these mice was greater than or equal to 11.1 mmol/L was identified as the diabetic model. The hyperglycemic mice were randomly divided into three groups: model group (MG, water), positive control group (PG, 10 mg/kg acarbose), low-dose polysaccharides group (LP, 100 mg/kg), and high-dose polysaccharides group (HP, 200 mg/kg). GLP and acarbose solution was given to mice by intragastric administration every day. All mice were provided with a conventional diet after modeling. Weight and FBG were monitored once a week for 4 weeks. The blood glucose level of the mice was measured from the tail veins by glucometer (Sinocare Inc., Changsha, China). In the final stage, blood was collected from the infraorbital angular vein after slight anesthetization. After blood collection, the mice were sacrificed for sampling the liver, kidney, and pancreas.

All procedures were performed in accordance with Public Health Service policies, the Animal Welfare Act, and the Laboratory Animal Committee (LAC) of South China University of Technology Policy on the Humane Care and Use of Vertebrate Animals. The ethic approval number is 2018001.

#### 3.5.2. Biochemistry Indexes Assessment

Serum was collected by centrifugation of blood at 4000 rpm for 15 min at 4 °C. Biochemical indexes including GSP, CRE, TG, TC, and T-AOC were measured on Multi-mode (PerkinElmer EnSpire, Singapore) by using assay kits.

The hepatic T-SOD and MDA were determined with the assay kit in accordance with the specification.

#### 3.5.3. Histopathological Examination

Liver, kidney, and pancreas tissues were embedded in paraffin. Paraffin sections were sliced and stained with hematoxylin–eosin (HE). The tissues were visualized using an CX41 microscope (Olympus, Japan) equipped with MDX4 (Mshot, Guangzhou, China) digital camera system under 100 × magnification.

### 3.6. Statistical Analysis

All experimental data were presented as means ± standard deviation (SD). Data were analyzed by one-way analysis of variance procedure with Duncan’s test (SPSS 17.0) (IBM, Chicago, IL, USA). *p* < 0.05 was considered as significant difference.

## 4. Conclusions

GLP exhibited excellent free radical scavenging activities in vitro. Furthermore, animal experiment results demonstrated that GLP exerted anti-diabetic effects in STZ-induced diabetic mice, significantly lowered FBG, TC, TG, GSP, CRE, MDA content, and increased T-AOC and T-SOD enzyme activity. Moreover, GLP could ameliorate liver, kidney, and pancreas damage. The overall findings suggest that polysaccharides from guava leaves could provide health benefits and might be considered as pharmaceutical or functional food ingredient.

## Figures and Tables

**Figure 1 molecules-24-01343-f001:**
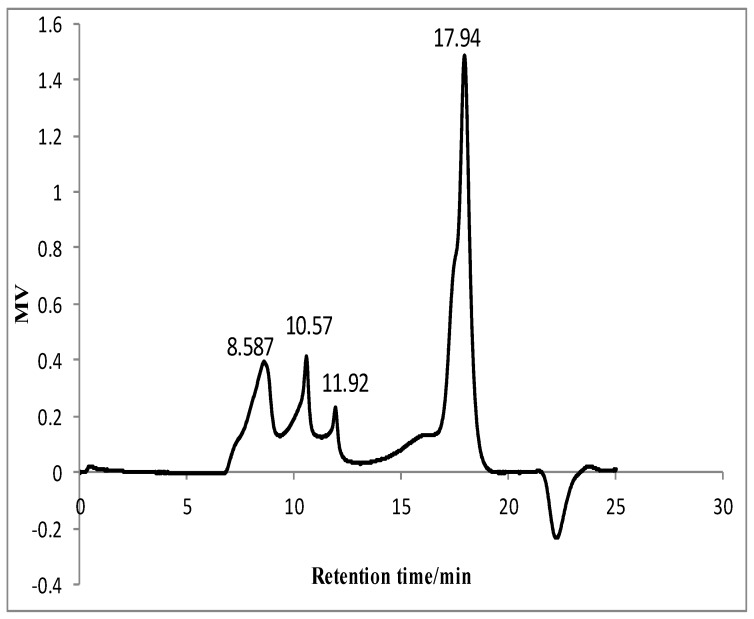
HPLC chromatograms of guava leave (GLP).

**Figure 2 molecules-24-01343-f002:**
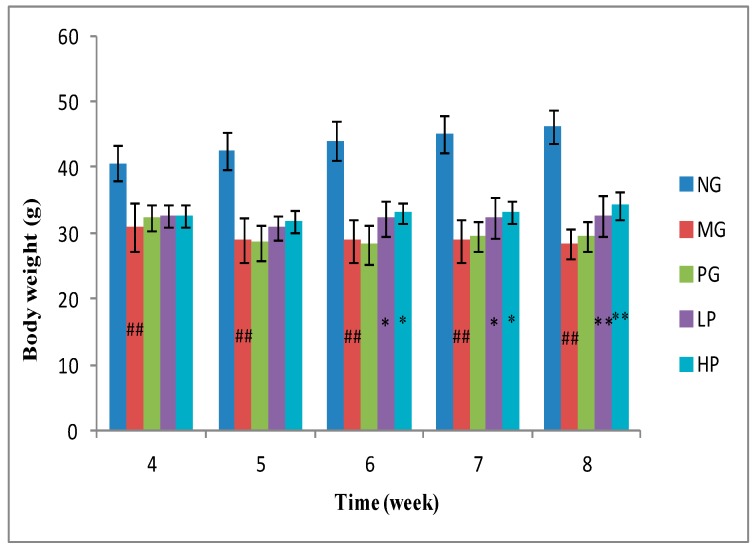
The body weight of mice in 5 weeks after modeling. NG, normal group; MG, model group; PG, positive group; LP, low-dose polysaccharides group; HP, high-dose polysaccharides group. “^##^” represents very significant difference compared with NG (*p* < 0.01); ** represents very significant difference compared with MG (*p* < 0.01); * represents significant difference compared with MG (*p* < 0.05).

**Figure 3 molecules-24-01343-f003:**
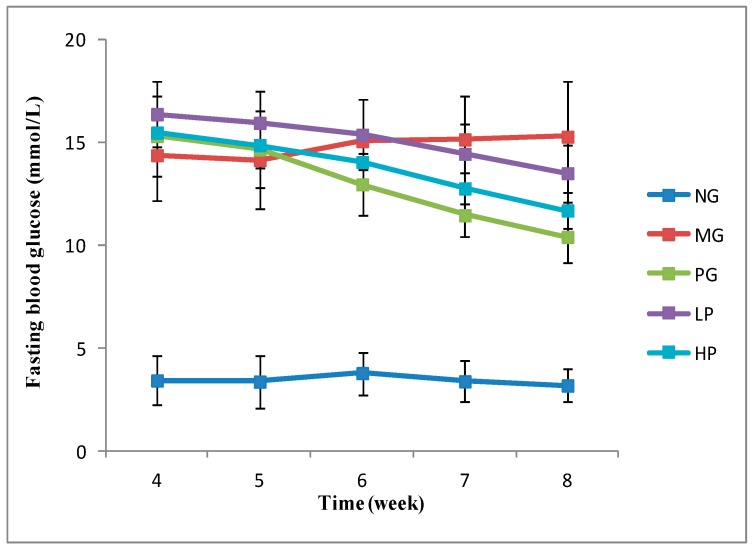
The fasting blood glucose of mice in 5 weeks after modeling.

**Figure 4 molecules-24-01343-f004:**
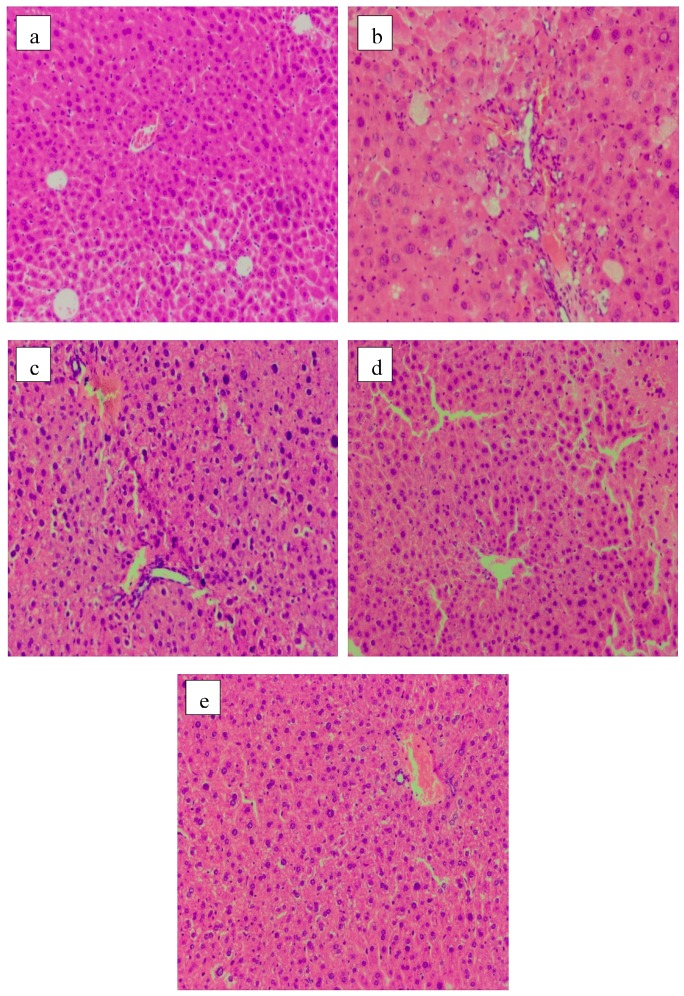
Liver histology images (HE staining, 100×). (**a**) NG; (**b**) MG; (**c**) PG; (**d**) LP; (**e**) HP.

**Figure 5 molecules-24-01343-f005:**
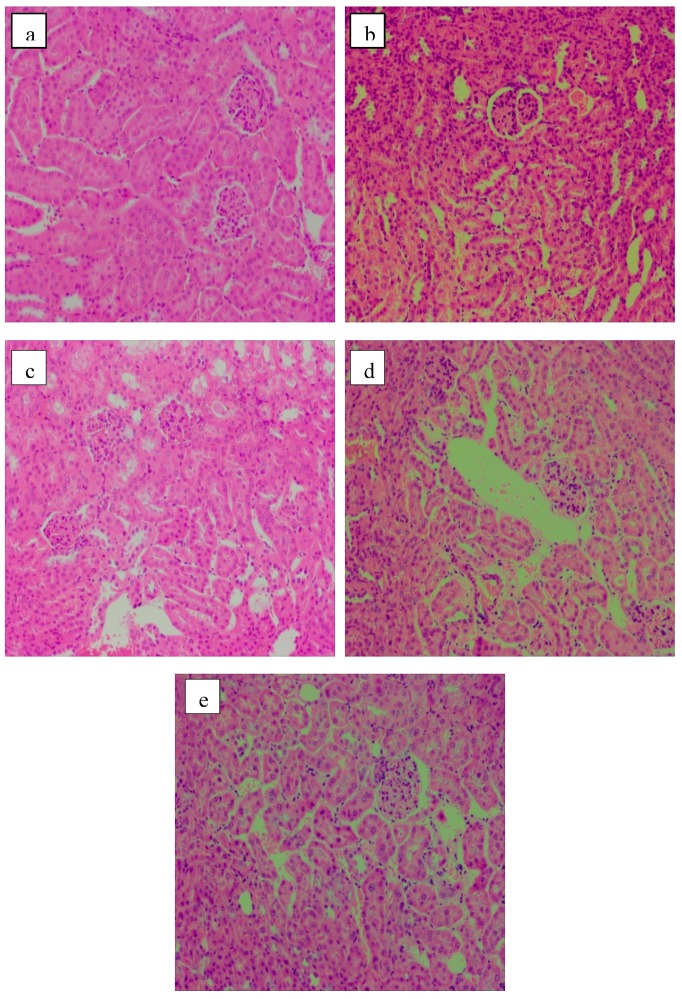
Kidney histology images (HE staining, 100×). (**a**) NG; (**b**) MG; (**c**) PG; (**d**) LP; (**e**) HP.

**Figure 6 molecules-24-01343-f006:**
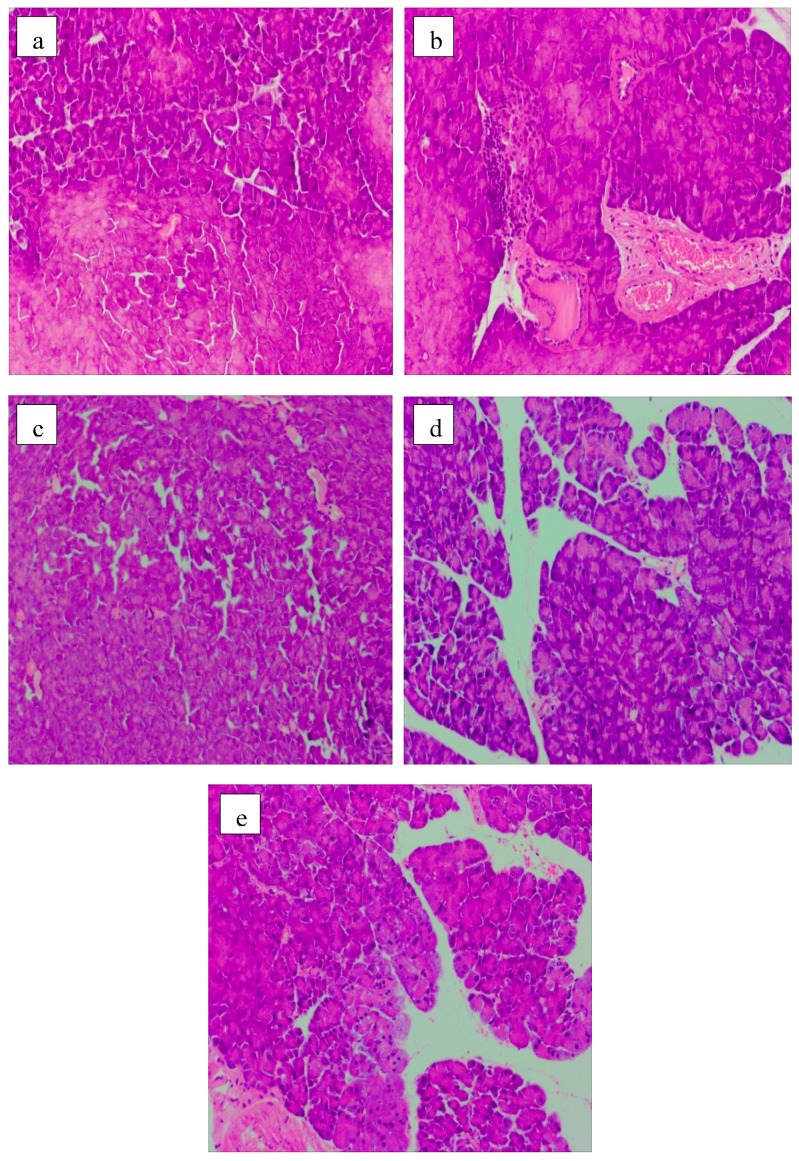
Pancreas histology images (HE staining, 100×). (**a**) NG; (**b**) MG; (**c**) PG; (**d**) LP; (**e**) HP.

**Table 1 molecules-24-01343-t001:** The antioxidant activities of GLP.

IC_50_/μg/mL	GLP	Positive Control
DPPH assay	46.49 ± 0.22	7.03 ± 0.15
OH assay	175.52 ± 0.31	119.37 ± 0.24
ABTS assay	102.82 ± 0.26	19.82 ± 0.11

**Table 2 molecules-24-01343-t002:** Regulation effects of polysaccharides on biochemical indicators.

Group	TC (mM)	TG (mM)	CRE (μM)	GSP (mM)	T-AOC (mM)	T-SOD (U/mgprot)	MDA (nmol/mgprot)
NG	2.71 ± 0.24	0.75 ± 0.14	9.43 ± 0.86	1.99 ± 0.11	0.84 ± 0.04	796.52 ± 17.35	6.94 ± 0.29
MG	6.71 ± 0.90 ^##^	2.18 ± 0.30 ^##^	22.52 ± 3.60 ^##^	3.07 ± 0.15 ^##^	0.55 ± 0.03 ^##^	608.95 ± 24.73 ^##^	12.50 ± 1.42 ^##^
PG	5.36 ± 0.51 **	1.76 ± 0.12 **	13.26 ± 1.21 **	2.48 ± 0.09 **	0.68 ± 0.02 **	721.83 ± 32.66 **	9.11 ± 0.35 **
LP	4.52 ± 0.63 **	1.35 ± 0.22 **	14.19 ± 1.14 **	2.85 ± 0.18 **	0.70 ± 0.01 **	702.13 ± 11.76 **	9.75 ± 0.37 **
HP	3.98 ± 0.35 **	1.32 ± 0.16 **	13.28 ± 1.04 **	2.43 ± 0.15 **	0.75 ± 0.02 **	730.80 ± 12.98 **	8.99 ± 0.26 **

^##^ represents very significant difference compared with NG (*p* < 0.01); ** represents very significant difference compared with MG (*p* < 0.01).

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
