# Peer review of "Antioxidant and Anti-Diabetic Activities of Polysaccharides from Guava Leaves"

_molecules, 2019, doi:10.3390/molecules24071343_

Round 1

Reviewer 1 Report

The manuscript (ID molecules-475523) displays the results of studies concerning the potential antioxidant (in vitro) and antidiabetic (in vivo) activities of polysaccharides from leaves of Psidium guajava L. The subject of interest seems to be important and well-corresponding to current scientific field of interest. The general comment is that the Authors used an “artificial” model in vivo. The polysaccharides were administered intragastrically so the salivary digestion was missed in this model.

On the other hand, it is difficult to find any correlation between the antioxidant properties of polysaccharides in vitro and “antioxidant” parameters in mice taking into consideration the bioavailability of polysaccharides or its lack. In this context the Authors completely missed any discussion. This needs any justification.

Minor comments

The full name of plant as well as plant family Psidium guajava L. (Myrtaceae) should be provided in the abstract.

L. 65-67 Please check the IC50 values, especially 119.72 μg/mL because in the Table 1 the IC50 175.52  μg/mL. In addition, these values are higher, not lower, than positive controls. Please add “…positive control, such as ascorbic acid or Trolox.”

L. 107-108 Please check the statement “Sugars in the blood cane be converted into triglicerides…”Does it take place in blood? Which sugars?

Table 2. Please unify the units mM=mmol/L

Author Response

1. The manuscript (ID molecules-475523) displays the results of studies concerning the potential antioxidant (in vitro) and antidiabetic (in vivo) activities of polysaccharides from leaves of Psidium guajava L. The subject of interest seems to be important and well-corresponding to current scientific field of interest. The general comment is that the Authors used an “artificial” model in vivo. The polysaccharides were administered intragastrically so the salivary digestion was missed in this model.

Response: Thanks very much for your suggestion. For liquid medicines, oral administration is usually administered by gavage in mice experiments. Xu et al. verified the anti-diabetic effects of polysaccharides from talinum triangulare in diabetic mice by intragastric administration (Xu et al., International Journal of Biological Macromolecules, 2015, 72, 575-579). Multiple toxicity studies of trehalose in mice were conducted by intragastric administration (Liu et al., Food Chemistry, 2013, 136:485-490).

2. On the other hand, it is difficult to find any correlation between the antioxidant properties of polysaccharides in vitro and “antioxidant” parameters in mice taking into consideration the bioavailability of polysaccharides or its lack. In this context the Authors completely missed any discussion. This needs any justification.

Response: Thanks very much for your valuable suggestion. Effects of saliva, gastric and intestinal digestion on the chemical properties, antioxidant activity of polysaccharides from guava leaves will be explored in our further study.

Free radical scavenging assays were performed to demonstrate the potential antioxidant activity of polysaccharides. Many studies verified that polysaccharides had good free radical scavenging ability in vitro and also exhibited good antioxidant activity in vivo. According to your suggestion, we added the discussion in the revised manuscript (line 140-145).

3.  The full name of plant as well as plant family Psidium guajava L. (Myrtaceae) should be provided in the abstract.

Response: Thanks very much for your suggestion. we have added the full name of plant and plant family Psidium guajava L. (Myrtaceae) in the abstract.

4. L. 65-67 Please check the IC50 values, especially 119.72 μg/mL because in the Table 1 the IC50 175.52  μg/mL. In addition, these values are higher, not lower, than positive controls. Please add “…positive control, such as ascorbic acid or Trolox.”

Response: Thanks very much for your suggestion. We feel sorry for our carelessness. We checked the IC50 values, and 119.72 μg/mL has been changed to 175.52  μg/mL. The word “lower” has been changed to “higher”. We have already added the sentence “…positive control, such as ascorbic acid or Trolox” in the revised manuscript (line 68).

5. L.107-108 Please check the statement “Sugars in the blood cane be converted into triglicerides…”Does it take place in blood? Which sugars?

Response: Thanks very much for your suggestion. Fat synthesis takes place in liver, adipose tissue and small intestine. Glycerol and fatty acids required for the synthesis of triglycerides are mainly supplied by glucose metabolism. Glycerol is converted from dihydroxyacetone phosphate produced by glycolysis, and fatty acids are synthesized from acetyl CoA produced by oxidative decomposition of sugar. The original expression is inappropriate, so we changed the sentence to “Triglyceride (TG) are the main constituents of body fat in humans…” in line 108.

6.  Table 2. Please unify the units mM=mmol/L

Response: Thanks very much for your suggestion. We have unified the units mM in Table 2.

Reviewer 2 Report

The manuscript entitled "Antioxidant and Anti-diabetic Activities of Polysaccharides from Guava Leaves"  reports results of a study about isolation of polysaccharides from guava leaves, and evaluation of antioxidant activity and anti-diabetes effects in diabetic mice. The manuscript can be accepted since brings some interesting results that can be used as a start point for further studies.  The text must be revised before, for instance, the difference of the letter size of the conclusion section and the rest of the manuscript.

Author Response

The manuscript entitled "Antioxidant and Anti-diabetic Activities of Polysaccharides from Guava Leaves"  reports results of a study about isolation of polysaccharides from guava leaves, and evaluation of antioxidant activity and anti-diabetes effects in diabetic mice. The manuscript can be accepted since brings some interesting results that can be used as a start point for further studies.  The text must be revised before, for instance, the difference of the letter size of the conclusion section and the rest of the manuscript.

Response: Thanks very much for your suggestion. The font and the letter size of the conclusion section have been modified.